# Development and Validation of a Questionnaire to Assess Adherence to the Healthy Food Pyramid in Spanish Adults

**DOI:** 10.3390/nu12061656

**Published:** 2020-06-03

**Authors:** Andrea Gila-Díaz, Silvia M. Arribas, Ángel Luis López de Pablo, Ma Rosario López-Giménez, Sophida Phuthong, David Ramiro-Cortijo

**Affiliations:** 1Department of Physiology, Faculty of Medicine, Universidad Autónoma de Madrid, 28029 Madrid, Spain; andrea.gila@uam.es (A.G.-D.); silvia.arribas@uam.es (S.M.A.); angel.lopezdepablo@uam.es (Á.L.L.d.P.); sophiph@kku.ac.th (S.P.); 2Department of Preventive Medicine, Public Health and Microbiology, Faculty of Medicine, Universidad Autónoma de Madrid, 28029 Madrid, Spain; mrosario.lopez@uam.es; 3Department of Physiology, Faculty of Medicine, Khon Kaen University, Khon Kaen 40002, Thailand; 4Division of Gastroenterology, Beth Israel Deaconess Medical Center, Harvard School of Medicine, 330 Brookline Avenue, Boston, MA 02215, USA

**Keywords:** diet, food intake, healthy food pyramid, nutrition, questionnaire design

## Abstract

We aimed to design and validate a new questionnaire of adherence to healthy food pyramid (HFP) (AP-Q), to improve previous instruments. The questionnaire was self-administered and included 28 questions from 10 categories (physical activity, health habits, hydration, grains, fruits, vegetables, oil type, dairy products, animal proteins, and snacks). A population of 130 Spanish adults answered it, obtaining scores from each category and a global score of HFP adherence (AP-Q score). Validation was performed through principal components analysis (PCA) and internal consistency by Cronbach’s alpha. AP-Q was also externally validated with Kidmed-test, answered by 45 individuals from the cohort. The global AP-Q score was 5.1 ± 1.3, with an internal consistency of 64%. The PCA analysis extracted seven principal components, which explained 68.5% of the variance. The global AP-Q score was positively associated with Kidmed-test score. Our data suggest that AP-Q is a complete and robust questionnaire to assess HFP adherence, with several advantages: easy to complete, cost-effective, timesaving and has the competency to assess, besides diet, several features affecting health status, lacking in other instruments. We suggest that AP-Q could be useful in epidemiological research, although it requires additional calibration to analyze its reproducibility and validation in other populations.

## 1. Introduction

Adequate nutrition provides the essential building blocks for growth, development and maintenance of a healthy status throughout life, being an important determinant of human health. The World Health Organization has identified unhealthy diets and physical inactivity, together with excessive alcohol and tobacco consumption, as the main modifiable risk factors contributing to non-communicable diseases [1]. Among them, dietary patterns play a major role. The role of the Mediterranean diet in the prevention of chronic diseases has been evidenced by numerous studies, including PREDIMED (Spanish acronym for Prevention with Mediterranean Diet). This study demonstrated the benefits of adherence to the Mediterranean diet, showing a 30% reduction in the incidence of major cardiovascular events and type 2 diabetes, and beneficial effects on metabolic syndrome, adiposity, cognition and breast cancer [2]. The Healthy Food Pyramid (HFP) was conceived by the Mediterranean Diet Foundation Expert Group, as a simplified graphical representation of the Mediterranean diet characteristics. It is the main framework for different socio-cultural contexts of the Mediterranean region, and aims to help adherence to this healthy dietary pattern [3].

Analysis of the population’s dietary patterns and degree of adherence to healthy habits provides information on the frequency and distribution of diets and/or nutritional status. However, dietary patterns are very difficult to measure, and inaccuracy of assessment may be a serious obstacle to understand its impact on disease and to design interventions at community level [4]. Therefore, obtaining reliable data on food consumption in individuals is a key factor and a necessary tool for health promotion and disease risk prevention [5]. Several instruments have been developed to measure dietary intake. Commonly used methods are the 24 or 72-h dietary recall method (24/72hDR), the food frequency questionnaire (FFQ), the duplicate diet approach, the dietary history, the dietary record, and the food consumption record [4,5,6]. Each of them has several advantages and disadvantages.

The 24hDR is a precise validated method for quantification of energy and nutrients, which estimates the usual intake of an individual in a short time, by an open-ended questionnaire administered by a trained interviewer [4]. Regarding the advantages, we can include the high response rate and the fact that it can be used with low literacy populations. However, this method involves an extensive reliance on the subject’s recent memory, depends on the ability of the interviewer to describe ingredients, food preparations, quantities or dishes. It also requires more than two 24hDR to estimate the usual intake, and it tends to underestimate consumption, especially in elderly and children [5,7].

The FFQ uses a predefined, self-administered format that collects the usual intake over a relatively long period (6 months or 1 year, for example). The advantage is that it is easy to assess the usual dietary intake, being cost-effective and timesaving. However, it must be specifically prepared for the study group and research aims, and its accuracy is low [7,8].

The duplicate diet approach assesses dietary intake by retaining a duplicate diet sample to collect information throughout a specific period and, therefore, it measures possible dietary exposures to certain components such as environmental contaminants. However, it is not suitable for large-scale studies [9].

The dietary history method is a subjective approach, which uses closed-ended questionnaires administered by a trained interviewer to estimate usual intake over a relatively long period. Its main disadvantages are the time taken to administer it and the high cost [10].

The dietary record is a self-administrated open-ended questionnaire, which provides information about intake during a specific period. It makes available detailed data without recall bias. However, it is lengthy, requiring literacy and highly motivated respondents. It is also expensive and time-consuming, since multiple days are required to evaluate accurately usual intake. In addition, possible changes in diet could happen if repeated measures are carried out [11].

The food consumption record is an objective observation of consumption, recorded by trained staff at household level. It is easy to use with lower literacy individuals or those who prepare most meals at home. However, it has low accuracy, and it is not suitable for those who eat outside frequently [4].

As stated above, the current dietary assessment methods have some pitfalls. To carry out an exhaustive, broad and complete analysis, it has been recommended to use a combination of several methods. Therefore, simpler, quicker, and cheaper methods to measure food intake are still needed. New methods must also be versatile and easy to adapt to new knowledge or dietary recommendations.

The present study aimed to design and validate an alternative manageable questionnaire, providing a general index of adherence to the HFP, which is the framework of the Mediterranean lifestyle, a healthy pattern being also used outside the Mediterranean region [12,13]. The Adherence to Pyramid questionnaire (AP-Q) that we propose combines open and open-ended questions, Likert scales, and frequency queries, to estimate the usual intake of all food categories over the last two months. To validate this questionnaire, we have evaluated the diet of 130 Spanish adults and their adherence to the recommendations of the HFP, revised by the Spanish Society of Community Nutrition in 2017 [14]. This questionnaire tries to improve previous instruments avoiding some limitations previously detected in studies exploring adherence to the Mediterranean diet [15]. In addition to diet questions, it includes other aspects affecting health status, such as physical activity, emotional balance, and self-perception of the health status and healthy habits, issues lacking in the rest of the instruments. The questionnaire is easy to fill, cost-effective and timesaving, since it does not require interviewer, being easy to modify if recommendations change. According to the results obtained, the AP-Q provides valid information about the degree of adherence to the HFP and can be a useful instrument for epidemiological studies.

## 2. Materials and Methods 

### 2.1. Study Design and Recruitment 

Adults who wished to participate in the project were recruited. They were informed in detail about the procedure and the objectives of the study, and then, signed the informed consent. All data obtained from the questionnaire were anonymously collected. This study was conducted according to the principles of the Declaration of Helsinki and was approved by the Ethics Committee of Universidad Autónoma de Madrid (Ref. CIE-102-1948). This was a pilot project and participants were recruited mainly through social networks and parenthood-specific discussion boards. The inclusion criteria were adults between 18 and 65 years old from both sexes, with Spanish language comprehension. The exclusion criteria were individuals with chronic diseases requiring pharmacological treatment, professional athletes or those following diets which restrict some foods, such as vegan or lacto-ovo vegetarian. The final cohort included 130 participants. 

### 2.2. Participants

Participants reported their age (years) and sex. Height was determined using a mobile stadiometer (KaWe person-check; Kirchner & Wilhemlm GmbH, Co. Asperg, Germany), with the subject’s head in the Frankfurt plane. Body weight was determined to the nearest 100 g using a digital scale (Omron, HBF-514C, IL, USA). Subjects were weighed in bare feet and light underwear, which was accounted for by subtracting 300 g from the measured weight. Body mass index (BMI) was calculated (kg/m^2^). The time taken to complete the AP-Q was also recorded.

### 2.3. AP-Q Instrument

AP-Q is a self-administrated questionnaire with 28 questions and multiple types of answers (Appendix A). The items included in the questions were distributed into 8 dimensions and 10 categories. It was developed systematically, using scientifically accepted methodology including, literature review, focus group discussions, expert evaluation and pre-testing [16]. A list of items was generated and represented AP-Q in a simple, lucid language. We took special care to provide proper sequencing and framing of the questions. The designed AP-Q draft was pre-tested in 20 individuals. Analysis for comprehensibility, replicability and ease of usage was analyzed during this phase, and irrelevant, ambiguous and duplicated questions were eliminated.

### 2.4. Proposed Categories and Dimensions in AP-Q Instrument

#### 2.4.1. Physical Activity

This category has three free-response questions referring to daily work related activity and usual sports practice. The maximum score is obtained with “moderate physical activity for at least one hour per day”.

#### 2.4.2. Healthy Habits and Culinary Techniques

This category has the following four dimensions: (1) Lifestyle: three binary-response questions “tobacco consumption”, “abuse drugs consumption” and “nutritional supplements consumption”. (2) Emotional balance: one question with 10 items, answered with a Likert scale from 0 to 3, where 0 represents “total disagreement” and 3 represents “total agreement” [17]. An example of this item is “The weight has great power over me”; higher scores indicate a lack of emotional balance related to nutrition. (3) Sleep hygiene: it assesses sleep frequency and satisfaction. It has an open-ended question on the number of sleep hours, and two multiple-choice questions on awakenings and time required to fall asleep. (4) Culinary techniques: it contains three multiple-choice questions, which analyze the number and distribution of daily intakes and cooking method, with special interest in the frequency of consumption of deep-fried foods, due to their implication in cardiovascular diseases.

#### 2.4.3. Hydration

This category has the following four dimensions. (1) Water intake: three free-answer questions, which assess the daily water intake, tea and/or coffee consumption, and how they are sweetened. (2) Soft drinks: one multiple-choice question assessing the frequency of soft drinks consumption. This dimension provides a negative score. (3) Wine and beers: one multiple-choice question including frequency of low alcohol beverages consumption. Non-alcoholic beers were considered as water intake since they provide hydration [18,19] and are a good source of polyphenols [20,21]. (4) Spirits (other alcoholic drinks): one binary-response question about the frequency of spirits intake. This score gives a negative score, penalizing intake of high alcohol content beverages.

#### 2.4.4. Grains, Seeds and Legumes

This category has one frequency answer with 17 items, including foods at the base of the HFP, such as legumes, starchy tubers, nuts, seeds, cereals and pseudo-cereals. This category will be named “grains” in the rest of the manuscript. We have included grains, seeds and legumes (including pulses) in the same category due to the presence of bioactive compounds and high-quality fats, which provide them with common functional properties against cardiometabolic diseases [22]. An update of the HFP includes whole grains and, therefore, in the calculation of this scale, their consumption scores positively.

#### 2.4.5. Fruits

This category has one frequency answer question with three items: consumption of whole fruits and juices, and commercial juices prepared out of concentrates. While fruit consumption provides a positive value, commercial juices score negative, because of their high sugar and low fruit content.

#### 2.4.6. Vegetables

This category has one frequency answer question with 12 items.

#### 2.4.7. Oil Type

This category has one free-response question. Extra virgin olive oil consumption provides a positive score, while consumption of spreadable fats, such as butter and margarine, is penalized. 

#### 2.4.8. Dairy Products

This category has one question with nine items assessing the frequency of milk, yogurt and cheese consumption. Consumption of semi-skimmed and low-fat dairy products is valued positively.

#### 2.4.9. Animal Proteins

This category has one frequency response with six items (classified in eggs, meat and fish), and one binary-response for seafood. Red and processed meats are penalized in the score, since it is known that excessive intake has a negative impact on health. Seafood consumption also gives a negative score, since seafood is currently missing in the HFP.

#### 2.4.10. Snacks

This category has one frequency answer question with 11 items. According to the HFP, intake of snacks should be moderate or occasional, since these foods have a low nutritional contribution, being in most cases rich in sugars, saturated fats, and salt. Therefore, their location in the HFP is in the summit. Accordingly, in the AP-Q their frequent consumption is negatively valued, except for occasional consumption of the items “dark chocolate” and “pickles”, which score positively. 

### 2.5. AP-Q Score

AP-Q score reflects the global adherence to the HFP. Therefore, the higher the score the better the adherence. Categories that refer to the bottom of the pyramid such as physical activity, healthy habits and some culinary techniques, hydration, grains, fruit or vegetables have positive scores, while categories referring to the top of the pyramid, such as snacks or to certain foods, that are not included in the pyramid, have negative scores (Appendix A). The minimum and maximum scores for each category are reported in Table 1.

To homogenize the contribution of each category to the AP-Q score, it was divided by its maximum limit, i.e., each category is adjusted in a range of −1 to 1. Those categories with a maximum of 0 were divided by their minimum limit. For categories that included several dimensions (“healthy habits and culinary techniques” and “hydration”), the scores were calculated by averaging their dimensions. The global AP-Q score was calculated as the sum of all the adjusted categories (adjusted AP-Q score). The minimum and maximum ranges of global adjusted AP-Q score were (−6; 10), and the unadjusted ranges were (−41.8; 84.4).

### 2.6. External Validation Procedure

To carry out an external validation of the AP-Q, 45 individuals from the cohort were also administered the Kidmed test [23,24]. Kidmed is a validated binary response test, including 16 items. From the sum of the values obtained in these items, the degree of adherence to the Mediterranean Diet was determined; the higher the score, the better the adherence. AP-Q and Kidmed tests were not administered simultaneously to avoid possible interference-bias between them.

### 2.7. Statistical Analysis

Statistical analysis was performed using IBM SPSS Statistical for Windows, version 25.0 (IBM Corporation, Armonk, NY, USA) software. The Kolmogorov–Smirnov test was used to evaluate the normal distribution of the variables. Quantitative normal variables were described as mean ± standard deviation (SD) and categorical variables as relative frequency. Correlations were tested by Rho-Spearman, and reliability was analyzed by Cronbach’s alpha. The suitability of the data for structure detection was reported by Kaiser–Meyer–Olkin (KMO). The KMO test indicates the proportion of variance in the variables, which may be caused by factors. A high value indicates that factor analysis may be useful. Bartlett’s test was used to report the sphericity. The Barlett’s null hypothesis tests if the variables are unrelated. Principal Components Analysis (PCA) was used to summarize information and those components with eigenvalue >1 were extracted. The simple structure of the main components was carried out by Oblimin. This is a rotation method that minimizes the saturation of the variables, simplifying their interpretation. Scores <0.2 were removed from the rotated matrix. A *p*-value <0.05 was established as statistical significance.

## 3. Results

### 3.1. Characteristics of the Participants

The sample consisted of 130 participants, equally distributed in both sexes (male = 43%; female = 57%), with an average age of 40.1 ± 15.1 years. The average weight was 70 ± 14.4 Kg and the height was 169 ± 9.4 cm. Average BMI = 24 ± 3.9 Kg/m^2^, with 92.5% of the participants below 30 kg/m^2^. The time taken to complete the AP-Q ranged between 10 and 20 min.

### 3.2. AP-Q Validation Approach: Internal Consistency Reliability

The adjusted global AP-Q score in the study was 5.1 ± 1.3, and the unadjusted score was 39.9 ± 11.8. Both scores were positively correlated (rho = 0.921; *p*-value < 0.001). The correlations between categories and dimensions of the AP-Q are shown in Table 2. In the cohort, AP-Q showed an internal consistency of 0.64. To validate the AP-Q, both the categories and dimensions of the questionnaire were considered. The KMO test was 0.74, with *p*-value < 0.001 for Barlett’s test. This indicated that the data variance could be caused by underlying factors, and PCA was performed, accordingly. The categories and dimensions of AP-Q were distributed in seven principal components (Figure 1A). Each component accounted for 22.1%, 11.0%, 8.8%, 8.2%, 6.6%, 6.1% and 5.8% of variance, respectively. Figure 1B shows the cluster that group in the three principal components with the most explained variance. 

In the first component, fruit and vegetable categories showed higher representation. In the second component, the variables related to hydration were the most relevant. The third component was mainly represented by the healthy habits’ category and sleep hygiene. The fourth component showed that the dairy products category was the most representative. In the fifth component, lifestyle was the dimension with the highest impact. In the sixth component, physical activity category was the most representative. In the last extracted seventh component, emotional balance was the dimension with the highest impact. The seven components explained 68.5% of the accumulative variance. The coefficients of the categories or dimensions in each component were extracted by oblique rotation (Oblimin) and the constants are shown in Table 3.

The projection of the categories and dimensions of AP-Q on the seven principal components allowed determining the weight and relationship of each component in the global AP-Q score (Table 4). The fifth component did not show a significant correlation with either the adjusted or the unadjusted global AP-Q score. However, the categories and dimensions summarized in the fifth component were also summarized in the other components.

### 3.3. Parallel-Forms Reliability between AP-Q, Kidmed and BMI

The Kidmed test was filled out by 45 individuals, due to sample loss. Kidmed test showed an average score of 7.8 ± 2.7. Kidmed score was statistically correlated with the adjusted global AP-Q score (rho = 0.670; *p*-value < 0.001) and the unadjusted AP-Q score (rho = 0.654; *p*-value < 0.001). However, BMI and Kidmed test did not show statistical correlation (rho = −0.118; *p*-value = 0.445). In our cohort, BMI did not exhibit statistical correlation neither with the adjusted AP-Q score (rho = −0.140; *p*-value = 0.130), nor with the unadjusted AP-Q score (rho = −0.170; *p*-value = 0.060).

## 4. Discussion

In this study we have designed and validated a new questionnaire (AP-Q) that measures the adherence to the HFP. Our results showed that the instrument, which was answered by 130 individuals, is easy and quick to complete. The sample was more heterogeneous and wider in age and global health status than previously validated questionnaires [11,25,26,27]. The heterogeneity of the population in the present study is a strength for validation, and suggests that AP-Q can be used with different age groups. If the sample is broadened in future studies, it could be segmented according to age, to obtain a typical score for each age. This segmentation would enable to establish a normative criterion of HFP adherence for a specific population, according to age. The AP-Q is not a diagnostic index, and obtained scores can only be considered as a degree of adaptation to the HFP. Likewise, the questionnaire categories were not designed for individuals who follow a strict diet for professional or medical reasons, such as high-performance athletes, those with diagnosed eating disorders or with an inflammatory/digestive pathology (ulcerative colitis, Crohn’s disease, inflammatory bowel disease, etc.), or individuals taking abuse substances. Future implementation of the AP-Q will allow using this questionnaire in different populations. This will require to redesign some of the questions. For example, those related to supplements in vegans, pregnant or breastfeeding women.

One of the advantages of the AP-Q as a method to estimate the quality of a population’s diet is the low-cost and simplicity. The main limitations of the most common direct methods (24 hDR or FFQ) to assess diet adherence when following a nutritional intervention, are their complexity and length. This results in consequent loss of participants, being difficult to apply in primary health care and preventive interventions [28,29,30,31,32,33]. In comparison, the AP-Q, developed in one instrument covering 10 crucial categories of diet and lifestyle, is an easy-to-use tool. In its present form, it was designed in paper format; however, AP-Q could be modified to create an electronic version to fill and correct online. As stated above, with the appropriate validation in specific populations, the simplicity of the proposed instrument will facilitate nutritional interviews, being useful in the clinical context. 

The AP-Q assesses different categories and dimensions incorporated in the HFP, including lifestyle, health habits, sleep, culinary techniques, and physical activity, which have not been evaluated in other questionnaires, which only focus on diet [15]. These categories are important to evaluate health status and the influence of diet patterns [34,35]. In fact, in our study, lifestyle showed a positive correlation with other categories, such as healthy habits, wine and beers and grains, and it was inversely correlated with water intake. The positive correlation between lifestyle and alcoholic drinks, such as wine and beer, could be related to the current controversy between the potential health benefits of polyphenols in wine versus the harmful effects of alcohol [36]. This controversy, together with the lack of knowledge in society, results in an increase in the rates of heavy drinkers [37,38]. There is a need for nutritional interventions that focus on reducing alcohol intake in those who consume it on a regular basis, rather than promoting consumption [39]. In this regard, education seems to play a key role in affecting adherence to healthy dietary habits [15].

On the other hand, culinary techniques had a positive correlation with healthy habits, water intake, hydration, grains, physical activity, oil type, snacks, and vegetables’ intake. The category of culinary techniques scores the different methodologies used in cooking. A high score in this category indicates a predominant use of healthy cooking techniques. These results support the relevance to include cooking techniques in questionnaires, as previously suggested [15].

Our data also evidenced other positive correlations between dietary patterns, such as vegetable and fruit intake associated with healthy lifestyle [40,41,42,43]. The evaluation of these categories, with AP-Q could guide diet education programs in the population.

The 10 categories and 8 dimensions were summarized in seven components by PCA, which explained over the 68% of the variability. Factor rotation is commonly used to improve the reliability and reproducibility of categories [44]. An inherent problem with this procedure is that there is not a unique solution to the rotation. The best solution is to decide a simple structure, which is achieved by rotating components around the origin categories until each component is co-linear with a distinct cluster of vectors [45]. An oblique rotation was used to extract the constant of each category over each component. The oblique rotation allows factors to be correlated. This rotation adds statistical complexity being more accurately represented, because constructs in the real world are rarely uncorrelated [45,46]. After the rotation, the weight of the categories and dimensions was distributed between the seven components, being all categories and dimensions included in some of them. However, physical activity, dairy products, and wine and beers, were only represented in the sixth, fourth and second components, respectively, with strong correlations. This result suggests that the second, fourth and sixth components would be essential to describe the diet behavior in the person.

As previously described, a high score in the model indicated good adherence to the HFP. The construction of the present questionnaire was designed with coherence to determine the adjustment to the HFP for everyone. The steps followed in the proposed analysis show coherence, since the associations between categories to generate the components follow a pattern with nutritional meaning. The correlation between the adjusted and unadjusted AP-Q scores corroborates the validation of the questionnaire to analyze HFP adherence. Both scores were also negatively correlated with the BMI of the participants, although there was no statistical significance. This may highlight that BMI is a variable that may be a poor indicator of a person’s overall health status [47].

In addition, the reliability of the AP-Q instrument was demonstrated by the external validation with the Kidmed test. This is a 16-items validated test which showed a positive correlation with the adjusted and unadjusted global AP-Q scores. Despite the fact that Kidmed is a quicker and simpler nutritional assessment tool [48], it does not explore in depth the individual’s nutritional pattern, while AP-Q, although longer, is a more elaborated and exhaustive questionnaire. Therefore, we suggest that the AP-Q offers more possibilities and it can be used to assess global HFP adherence (including all categories and dimensions), or to evaluate adherence to a specific part of the HFP (choosing a particular category or dimension).

### Study Limitations

The present study has some limitations. The number of participants who answered the final draft questionnaire is low, and a larger sample may reveal further valid clusters. Moreover, it would have been interesting to collect consumption of plant-based beverages, dried fruits or jams and salty snacks, in order to make the questionnaire broader. In its present form, AP-Q does not include them since they are not part of the HFP. A third aspect is the combination of several different foods into the category “grains”, which was decided based on their content in bioactive compounds (such as antioxidants) and high-quality fats, which confer them with functional properties against cardiometabolic diseases [22]. However, since the different foods included under this category differ in the content of several nutrients, future AP-Q versions could benefit from the separation of the grain’s category into the dimensions of cereals and cereal products, nuts and seeds, and legumes, extracting a particular score from each of them. Another limitation is the influence of recall and perception in the answers of the participants, particularly regarding “sleep hygiene” and “emotional balance. This is an inherent problem common in questionnaire methodology. In the study, participants had one month to answer the AP-Q, and it is possible that reducing this period, can minimize this limitation. 

## 5. Conclusions

The AP-Q instrument is a new questionnaire, which measures the adherence of diet patterns to the recommendations of the HFP in adults. It combines the strengths of the most commonly used methods to record food intake. Furthermore, the inclusion of additional information such as lifestyle, physical activity, sleep hygiene and emotional balance, makes this instrument more holistic to detect behavioral changes during nutritional interventions. Global AP-Q score is related to Kidmed test score, providing external validation. Although the metric properties of the AP-Q categories and dimensions are sturdy, additional studies of reproducibility and validity would help to standardize the proposed scores. We expect that AP-Q will be useful in epidemiological research.

## Figures and Tables

**Figure 1 nutrients-12-01656-f001:**
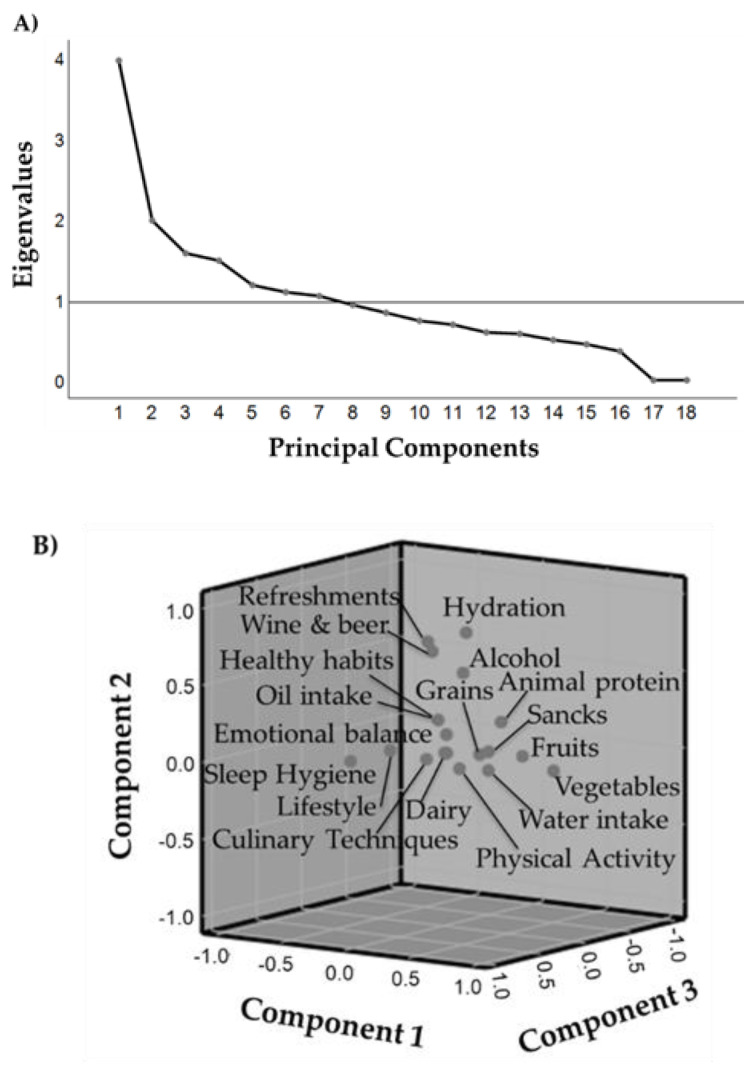
Eigenvalue (**A**) and scatter plot (**B**) of the distribution of the principal components for AP-Q categories and dimensions. The plot line shows eigenvalue = 1. The scatter plot shows the three first components.

**Table 1 nutrients-12-01656-t001:** Categories and dimensions proposed in the adherence to healthy food pyramid (AP-Q).

Category	Dimension	(Max; Min)
Physical activity		(0; 21)
Healthy habits and culinary techniques	Lifestyle	(−9; 5)
	Emotional balance	(0; 3)
	Sleep hygiene	(−3; 13)
	Culinary techniques	(−8; 14)
Hydration	Water intake	(−4.5; 7.5)
	Soft drinks	(−7.2; 0)
	Wine and Beers	(−4.6; 0)
	Spirit drinks	(−14; 0)
Grains, seed and legumes		(0; 11.3)
Fruits		(−3; 5)
Vegetables		(0; 5)
Oil type		(−2; 6)
Dairy products		(0; 9)
Animal protein		(−0.7; 8)
Snacks		(−0.8; 2.8)

**Table 2 nutrients-12-01656-t002:** Correlations between the categories and dimensions of AP-Q.

	1	2	3	4	5	6	7	8	9	10	11	12	13	14	15	16	17	18
1. Physical Act.	7.1 ± 10.1																	
2. Lifestyle	−0.17	2.9 ± 2.0																
3. Emotional bal.	−0.04	0.02	0.7 ± 2.0															
4. Sleep Hygiene	0.01	0.09	0.14	2.9 ± 8.4														
5. Culinary tech.	0.20 *	0.02	−0.16	0.11	3.8 ± 6.8													
6. Healthy habits	0.02	0.50 *	0.08	0.59 *	0.69 *	1.5 ± 4.8												
7. Water intake	0.14	−0.17 *	−0.08	0.02	0.37 *	0.17	2 ± 3.8											
8. Soft drinks	0.09	0.15	0.15	0.13	0.15	0.23 *	−0.03	2.2 ± 2.6										
9. Wine & Beers	−0.09	0.28 *	0.09	0.24 *	0.05	0.26 *	-0.01	0.28 *	0.5 ± 0.6									
10. Spirit drinks	0.04	0.09	−0.06	0.05	0.09	0.13	0.09	0.06	0.26 *	1.9 ± 0.8								
11. Hydration	0.14	0.12	0.04	0.12	0.36 *	0.34 *	0.53 *	0.71 *	0.35 *	0.38 *	3.8 ± 0.3							
12. Grains	0.29 *	0.28 *	−0.12	0.01	0.31 *	0.28 *	0.26 *	0.04	0.09	0.16	0.25 *	2.2 ± 4.5						
13. Fruits	−0.02	0.14	0.12	−0.07	0.05	0.09	0.13	0.05	0.02	0.02	0.15	0.32 *	1.7 ± 3.8					
14. Vegetables	0.06	0.16	0.03	−0.08	0.17 *	0.17 *	0.33 *	0.02	−0.03	0.15	0.21 *	0.34 *	0.40 *	0.9 ± 3.0				
15. Oil type	−0.00	0.00	0.15	0.17	0.23 *	0.25 *	0.15	0.13	0.22 *	0.13	0.25 *	0.11	0.13	0.18 *	1.9 ± 3.9			
16. Dairy products	0.05	−0.00	−0.6	−0.05	0.12	0.02	0.09	0.11	−0.09	0.12	0.20 *	0.27 *	0.19 *	0.09	0.08	1.0 ± 3.9		
17. Animal protein	0.07	0.13	0.02	0.06	0.15	0.18 *	0.16	0.18 *	0.12	0.12	0.30 *	0.22 *	0.16	0.28 *	0.14	0.05	1.1 ± 4.4	
18. Snacks	0.31 *	−0.02	0.18 *	0.22 *	0.34 *	0.35 *	0.32 *	0.17	0.07	0.24 *	0.35 *	0.36 *	0.20 *	0.30 *	0.18 *	0.08	0.13	0.4 ± 1.9

* *p*-value < 0.05; Rho–Spearman correlations; n = 130. Diagonal shows the mean ± SD. Activity (Act.); balance (bal.); techniques (tech.).

**Table 3 nutrients-12-01656-t003:** Coefficients of categories or dimensions on the principal components with oblique rotation.

	Principal Component						
	1st	2nd	3rd	4th	5th	6th	7th
Physical Activity						0.76	
Healthy Habits	0.21		0.90		0.31		
Hydration	0.31	0.85					
Grains	0.48			0.46	0.30	0.31	−0.22
Fruit	0.61			0.35		−0.22	0.26
Vegetables	0.82						
Oil Type		0.29	0.31		-0.31		0.22
Dairy Products				0.81			
Animal Protein	0.46	0.26		−0.22		0.29	
Snack	0.52		0.34			0.23	
Lifestyle			0.24		0.82		
Emotional Balance							0.86
Sleep Hygiene	−0.21		0.74				0.28
Culinary Techniques	0.33		0.66				−0.37
Water Intake	0.54		0.27		−0.51		−0.29
Soft drinks		0.74				0.46	
Wine and Beer		0.68					
Spirit Drinks		0.55		0.46		−0.34	

**Table 4 nutrients-12-01656-t004:** Correlations between principal component analysis (PCA) and Global AP-Q Score (GS AP-Q).

	1st PC	2nd PC	3rd PC	4th PC	5th PC	6th PC	7th PC	^1^GS AP-Q	^2^GS AP-Q
**1st PC**	13.34 ± 4.5								
**2nd PC**	0.60 *	−0.75 ± 5.4							
**3rd PC**	0.64 *	0.41 *	19.46 ± 5.5						
**4th PC**	0.67 *	0.32 *	0.27 *	5.19 ± 2.1					
**5th PC**	0.09	0.03	0.21 *	0.26 *	1.35 ± 3.13				
**6th PC**	0.29 *	0.22 *	0.15	0.18 *	−0.06	9.06 ± 5.9			
**7th PC**	−0.60 *	−0.16	−0.35 *	−0.33 *	0.10	−0.25 *	0.36 ± 2.2		
**^1^** **GS AP-Q**	0.88 *	0.69 *	0.62 *	0.67 *	0.10	0.48 *	−0.31 *	5.06 ± 1.3	
**^2^** **GS AP-Q**	0.76 *	0.59 *	0.52 *	0.60 *	0.07	0.74 *	−0.36 *	0.92 *	39.87 ± 11.8

Principal component (PC); ^1^ adjusted; ^2^ unadjusted. * *p*-value < 0.05; Rho–Spearman correlations. The diagonal shown the mean ± SD.

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
