# Peer review of "Development and Validation of a Questionnaire to Assess Adherence to the Healthy Food Pyramid in Spanish Adults"

_nutrients, 2020, doi:10.3390/nu12061656_

Round 1

Reviewer 1 Report

This is a new questionnaire for the measurement of the adherence to Healthy 

Food Pyramid. But some of the like " Emotional balance", "Sleep hygiene" may be low accuracy and too rough. The intake frequency of fruits, vegetables and grains, animal protein and snacks may also depends on subject's memory.  Therefore we don't know if this is a robust questionnaire to assess adherence to the Healthy Food Pyramid.

Author Response

We would like to thank the reviewer for the comments and suggestions.

We have considered in our questionnaire, multiple areas of the nutrition behaviors of the individuals, included in the healthy food pyramid, that can be evaluated together. We agree with the reviewer that certain categories, such as “Emotional balance” (which is already an adapted and validate in Spanish population test of Carol Ryff's Psychological Well-Being Scales; PMID: 17296089) or “Sleep hygiene” should be deepened. However, this questionnaire is one of the first tools that evaluates in a holistic way the nutritional behavior of the individual. Regarding the concern about memory, it is a general problem with questionnaires which should always take into account. We have included this point in the background section. We believe that we have collected sufficient information in the result section to consider a suitable tool. Nevertheless, we are aware of some limitations and we think the questionnaire can be improved and further validates in other populations.  

Reviewer 2 Report

The manuscript is well written and the method well described. Bibliographic references are appropriate to the topic. I recommend some minor revisions that do not significantly affect the quality of the work:
-In the introduction, many parts are taken from reference 4. I recommend finding other references to complete the part or to slightly shorten this section.
-About food frequency questionnaire tool, I recommend adding the commonly accepted abbreviation FFQ.
-The inclusion criteria specify that any vegan individuals have been excluded. Does this also apply to ovo-lacto vegetarians or have they been included?
-The "grains" category in the manuscript also contains legumes, seeds and starchy tubers. Is it possible to find a different definition of the group that is more precise?
-It is not displayed the total number of individuals selected for external validation by Kidmed test, moreover, it is not specified with which criterion this subgroup was selected.
-There are typos (extra space) at lines 200 and 283
-In the results section, tables 2 and 3 should be indicated in the text following their descriptions
-The sentence at lines 282-284 should be reworded to be clearer
-In the references list, pay attention to the journal references formatting

Author Response

We would like to thank the reviewer for the comments and suggestions. Bellow, we have answered point-by-point kindly your suggestions.

  • In the introduction, many parts are taken from reference 4. I recommend finding other references to complete the part or to slightly shorten this section.

Answer: According your recommendations, we have added some more references in the introduction (References: 8-11).

  • About food frequency questionnaire tool, I recommend adding the commonly accepted abbreviation FFQ.

Answer: We have included the suggested abbreviation for food frequency questionnaire from line 55 onwards.

  • The inclusion criteria specify that any vegan individuals have been excluded. Does this also apply to ovo-lacto vegetarians or have they been included?

Answer: Yes, ovo-lacto-vegetarians were also excluded, since they were a group with dietary restrictions. The sentence was modified to clarify the readers (lines 115-116).

  • The "grains" category in the manuscript also contains legumes, seeds and starchy tubers. Is it possible to find a different definition of the group that is more precise?

Answer: We agree with reviewer´s suggestion and we have now defined the 4th category as “Grains, seeds and legumes”, which included legumes, starchy tubers, nuts, seeds, cereals and pseudocereals (lines 157-161).

  • It is not displayed the total number of individuals selected for external validation by Kidmed test, moreover, it is not specified with which criterion this subgroup was selected.

Answer: The AP-Q and the Kidmed test were temporarily separated to avoid possible interferences-bias between them. We provided the AP-Q, did an intermediate statistical analysis to establish its internal validation and consistence of AP-Q and thereafter, the external validation by Kidmed test was provided to the participants. Unfortunately, we had loss of follow-up by SARS-COV-2 pandemic lockdown, which made it difficult to collect the sample and Kidmed test was performed in 45 individuals. However, the results of Kidmed already confirm the results obtained by internal validation of AP-Q. We have included the information in lines 206-207.

  • There are typos (extra space) at lines 200 and 283

Answer: Thank you for the detection of typos. We hope they have been all corrected.

  • In the results section, tables 2 and 3 should be indicated in the text following their descriptions.

Answer: Thank you. We have tried to modify table 2. Unfortunately, we could not edit the manuscript to place the tables following the text (journal requirements).

  • The sentence at lines 282-284 should be reworded to be clearer.

Answer: According to the reviewer comments, the sentence has been re-written; we hope the meaning is now clearer (currently, lines 296-298).

  • In the references list, pay attention to the journal references formatting

Answer: The references were edited according to Nutrients style.

Reviewer 3 Report

The manuscripts presents the validation of an alternative questionnarire for evaluating adult dietary patterns although authors already suggest it may have no value for diagnosis. The questionnaire should have been included to be able to read the questions (The Appendix A with the AP-Q corrections seems not enough to review the development and validation that authors are trying to publish)

The novelty of the questionnaire is based on the inclusion of variables related to lifestyle, physical activity, sleep and others like culinary techniques and emotional balance. Nevertheless, there are 3 questions about sleep and only one question (Q7) related to emotional intake according to the Appendix A).

It will be a low cost tool that should be modified into and electronic version as the authors suggest in line 290. The design of the study is excellent and the struture of the text is also good. Tables and references are easy to read and updated.

Bellow there are a few comments to the authors:

Abstract mentions 10 categories in line 17 and then in line 20 the list of them but there are only 8. Are these dimensions or categories?. Later on, line 125 refers to 8 dimensions.

Line 113: exclusion criteria: patients under pharmacological treatment were excluded from the study. Why those that intake "nutritional supplements" (line 138) were not excluded?. What do authors consider a nutritional supplement?

Line 137: "use of drugs" means "use of abuse drugs". Could the use of the trem "drug" generate a misunderstanting with terapeutical agents? 

Line 220: the sample size was 130: was this the final number of participants that filled the questionnaire?. Why many lost participants did you have?. What was the original number of participants that received the questionaire? The authors mention something about this limitation in line 355 but without giving the numbers.

Appendix A shows the AP-Q correction but why do not the authors add the questionnair it self?

Line 392: Why are students classifies with passive professions and evaluated with 0?

lines 405 and 407 mention and "etc" that does not seem to be very scientific. What are those other situations that could be included in that "etc"?

Line 443: What kind of refresments are included or considered in this dimensions?. I guess we all agree there are many types of refresments and most of them with different health value or nutrional value. Would a non alcholoic beer be considered a refresment inseat of water? It makes no sense to considere non alcoholic beer as water (line 452). What is a refresment for the adult Spanish population? In case the authors wanted to validate this questionnaire your young adults would this dimension and others need to be re designed?

Line 398: Why lifestyles in plural?

Where is the dried fruits consumption considered? And Legumes?

In the conclusions authors point out that reproducibility and validity need of additional studies. So why including "validation in the tittle" and not only just "development"

Author Response

We would like to thank the reviewer for the comments and suggestions. Bellow, we have answered point-by-point kindly your suggestions.

  • Abstract mentions 10 categories in line 17 and then in line 20 the list of them but there are only 8. Are these dimensions or categories? Later on, line 125 refers to 8 dimensions.

Answer: The AP-Q has 10 categories and 8 dimensions. The dimensions are only included in 2 categories, (4 dimensions in the category “Healthy habits and culinary techniques” and 4 dimensions in the category “Hydration”). The most inclusive cluster is category. The categories are included in the abstract and the organization of categories and dimensions is depicted in Table 1. The abstract has also been amended for clarity

  • Line 113: exclusion criteria: patients under pharmacological treatment were excluded from the study. Why those that intake "nutritional supplements" (line 138) were not excluded? What do authors consider a nutritional supplement?

Answer: We excluded patients since we thought it was better to validate this questionnaire, initially in a healthy population without the potential interference of drugs, which could affect appetite, sleep or physical activity (i.e. the antidiabetic metformin decreases appetite or the antidepressant imipramine produces somnolence). However, since we expect that this questionnaire could be used in the future in studies involving patients, we have included a question about the use of regular medication (Q5).

Regarding nutritional supplements, we consider them a dietary complement (usually including, vitamins, minerals and some essential fatty acids). They are frequently used in the population without medical control and, in a well-nourished population, they are not needed. In fact, supplements are not included in the healthy food pyramid. Since our aim was to assess the adherence to the pyramid, their intake is penalized in the AP-Q. However, if this questionnaire is later on used for other populations, which require the use of supplements (i.e. pregnancy, lactation), it should not be penalized. To clarify this aspect, we have included a note in the correction of Q5 in Appendix B: “future validation of the questionnaire should consider this item as non-penalty if there are special diets (vegans), situations (pregnancy, lactation) or diseases, which require supplementation”. This is also included in the discussion (Lines 300-302).

  • Line 137: "use of drugs" means "use of abuse drugs". Could the use of the term "drug" generate a misunderstanding with terapeutical agents?

Answer: We agree with this and we have changed to “abuse drugs consumption” (Line 139).

  • Line 220: the sample size was 130: was this the final number of participants that filled the questionnaire? Why many lost participants did you have? What was the original number of participants that received the questionnaire? The authors mention something about this limitation in line 355 but without giving the numbers.

Answer: The entire cohort was 130 participants and we did not have any loss in the answer of the AP-Q. The Kidmed test was answered by 45 participants. The AP-Q and the Kidmed test were temporarily separated to avoid possible interferences-bias between them. We provided the AP-Q, did an intermediate statistical analysis to establish its internal validation and consistence of AP-Q and thereafter, the external validation by Kidmed test was provided to the participants. Unfortunately, we had loss of follow-up by SARS-COV-2 pandemic lockdown, which made it difficult to collect the sample and Kidmed test was performed in 45 individuals. We have included the information in lines 206-207.

  • Appendix A shows the AP-Q correction but why do not the authors add the questionnaire itself?

Answer: We agree with the reviewer and we have included the questionnaire in the Appendix A and the corrections in Appendix B (no supplementary material in the revised version).

  • Line 392: Why are students classifies with passive professions and evaluated with 0?

Answer: The Q2 refers to additional daily physical activity (not included in Q1), performed by persons with professions which involve a physical work (i.e. is not a sedentary profession). In the Spanish context, students do not often carry out a work activity and would score 0 in this question (as a passive workers). The same applies to retired persons, for example. We have substituted the sentence in Q2 “For students and passive professions, the score is 0” to “For professionals with passive works or individuals with no job the score is 0” (Lines 498-499).

However, it a student has a job involving a physical activity, it would score 1. Q2 does not include walking go the university or educational center, which is evaluated in Q3.

  • lines 405 and 407 mention and "etc" that does not seem to be very scientific. What are those other situations that could be included in that "etc"?

Answer: We agree with the reviewer comment and we have removed the word “etc”.

  • Line 443: What kind of refreshments are included or considered in this dimensions? I guess we all agree there are many types of refreshments and most of them with different health value or nutritional value. Would a non-alcoholic beer be considered a refreshment inseat of water? It makes no sense to consider non alcoholic beer as water (line 452). What is a refreshment for the adult Spanish population? In case the authors wanted to validate this questionnaire your young adults would this dimension and others need to be re designed?

Answer: By refreshments we mean soft drinks (drinks with sugar, light, zero, energy drinks) as stated in the Q18. We have renamed the dimension to clarify to the readers. None of them are recommended in the healthy food pyramid, and for this reason, they score negatively.

Regarding question about non-alcoholic beer, we had a debate on this aspect. Finally, we decided to include it together with water since non-alcoholic beer has health benefits, due to its polyphenol content (PMID: 23186386; PMID: 23484672) and it is regarded as part of the Mediterranean diet and rehydration process (PMID: 27800480; PMID: 26056515). We have included this information in lines 154-155.

  • Line 398: Why lifestyles in plural?

Answer: You are right. We have modified the word.

  • Where is the dried fruits consumption considered? And Legumes?

Answer: We agree with the reviewer comment, because there is not particularly item related to dried fruits. This consideration could be added to improve the AP-Q in futures versions and we included this aspect in the line 371. In the other hand, Legumes are included in the “grains” and the category IV has been modified (line 157).

  • In the conclusions authors point out that reproducibility and validity need of additional studies. So why including "validation in the tittle" and not only just "development"

Answer: We consider that the present study is not only the development, but also a validation, since it includes a comparison with an established test (Kidmed) and factor analysis. However, all behavioral measurements require caution in their conclusions and we are aware of some limitations and further validation would be desirable. For example, expanding expand the sample size to determine Z or standardized scores would allow the comparison between populations with different nutritional patterns (i.e. Spanish and Thai). We have adapted our conclusion in line 381.

Reviewer 4 Report

This is an origina manuscript, of some interest.

It is not easy to find the dataset size, this should be given in the Abstract, Methods and Table 2, instead that in the first line of the Results only.

The methods are relativrly complex for a nutrition journal, and the paprer my therefore be not easy to understand for several readers.

Author Response

We would like to thank the reviewer for the comments and suggestions. Bellow, we have answered point-by-point kindly your suggestions.

  • It is not easy to find the dataset size, this should be given in the Abstract, Methods and Table 2, instead that in the first line of the Results only.

Answer: According to reviewer suggestion, we have added the sample size information in the abstract, methods and table 2.

  • The methods are relatively complex for a nutrition journal, and the paper my therefore be not easy to understand for several readers.

Answer: We are aware of the complexity of the methodology, which may seem to be slightly out the general scope of a nutritional journal. However, we consider essential the thorough description, since our principal aim was to validate a new questionnaire. For this, we think it is necessary to describe the methods in detail, and to perform a robust statistical analysis. We expect that the manuscript fits in the special issue “Dietary Assessment in Human Health and Disease” and we hope that the proposed questionnaire can be useful in this context.

Round 2

Reviewer 1 Report

Although the questionnaire is  easy to complete, cost effective, but I think some item still not very accurate and depends on subject's memory (sleep, and emotional balance).  

Author Response

Answer: the authors want to thank the review’s comment. We agree with you on this limitation. All questionnaires, as indirect measurements of human behavior, have in common the problems of recall and perception. Possibly, this would affect to a larger extent sleep and emotional balance. We have now highlighted this aspect in the manuscript as a limitation to be taken into account; we think this can be improved by reducing the time the subject has to recall to answer the questionnaire (lines 382-386).